# Pharmacological Approach to Sleep Disturbances in Autism Spectrum Disorders with Psychiatric Comorbidities: A Literature Review

**DOI:** 10.3390/medsci6040095

**Published:** 2018-10-25

**Authors:** Sachin Relia, Vijayabharathi Ekambaram

**Affiliations:** 1Department of Psychiatry, University of Tennessee Health Sciences Center, 920, Madison Avenue, Suite 200, Memphis, TN 38105, USA; 2Department of Psychiatry, University of Oklahoma Health Sciences Center, 920, Stanton L Young Blvd, Oklahoma City, OK 73104, USA

**Keywords:** autism spectrum disorder, sleep disorders in ASD, medications for sleep disorders in ASD, comorbidities in ASD

## Abstract

Autism is a developmental disability that can cause significant emotional, social and behavioral dysfunction. Sleep disorders co-occur in approximately half of the patients with autism spectrum disorder (ASD). Sleep problems in individuals with ASD have also been associated with poor social interaction, increased stereotypy, problems in communication, and overall autistic behavior. Behavioral interventions are considered a primary modality of treatment. There is limited evidence for psychopharmacological treatments in autism; however, these are frequently prescribed. Melatonin, antipsychotics, antidepressants, and α agonists have generally been used with melatonin, having a relatively large body of evidence. Further research and information are needed to guide and individualize treatment for this population group.

## 1. Introduction

Autism is a developmental disability that can cause significant emotional, social, and behavioral dysfunction. According to the Diagnostic and Statistical Manual (DSM-V) classification [1], autism spectrum disorder (ASD) is characterized by persistent deficits in domains of social communication, social interaction, restricted and repetitive patterns of behavior, interests, or activities. The term autism spectrum disorder has replaced the earlier terminology of pervasive developmental disorder in the classification systems of DSM-IV TR [2]. There is greater variability in clinical presentation of ASD, depending upon each individual’s intelligence quotient (IQ) level. Intelligence quotient is a major determinant in determining the degree of impairment among individuals with ASD. In DSM-V, ASD is categorized as low-functioning autism with below average intelligence quotient (<70) and high-functioning autism with above average intelligence quotient (≥70) [1]. 

Studies have documented individuals with low-functioning autism experience significant impairments in their ability to function and exhibit serious behavioral disturbances, self-injurious behaviors, and socially inappropriate behaviors, and these individuals have higher predisposition to sleep–wake disturbances compared to their counterparts with high-functioning autism [3,4,5]. Overall, it is estimated that about 1 in 59 children suffer from ASD [6]. Sleep disorders occur frequently in autism spectrum disorder, with some studies reporting a prevalence between 50%–80% [7].

Studies have suggested low-functioning autism with increased severity, such as language deficits, increases the likelihood of sleep problems and can worsen the severity of sleep problems in these individuals [8,9]. There is also a lack of available data in low-functioning autism because of challenges in obtaining actigraphy and polysomnography studies in these members of the population [10].

Various potential mechanisms, including delayed melatonin peak, reduced rhythm amplitude, reduced ferritin, and increased periodic limb movements in sleep, have been hypothesized as a cause of sleep problems in ASD [11,12,13]. Sleep problems in individuals with ASD have also been associated with poor social interaction, increased stereotypy, problems in communication, and overall autistic behavior [14].

It is well established that psychiatric morbidities frequently co-occur in autism spectrum disorders and play a major role in sleep dysregulation. Conditions like anxiety and attention-deficit hyperactivity disorder (ADHD) in the ASD population can increase arousals, delay sleep onset latencies, and contribute to insomnia. Seizures and gastrointestinal disorders (like diarrhea) can cause frequent awakenings, sleep deprivation, and disruption in the sleep cycle [15].

Though sleep problems account for the most common reason for medications being prescribed, even in younger children, it is interesting to note that no medication is U.S. Food and Drug Administration (FDA)-approved for treatment of sleep disorders in children. In a community survey, approximately 75% of practitioners had recommended nonprescription medications and about 50% had prescribed a sleep medication during a six-month practice period [16].

Generally, behavioral interventions should be considered primary modalities of treatment and should be initiated before considering pharmacotherapy. The functioning level of autism (high- vs. low-functioning) should be taken into consideration before determining treatment approaches. The basic tenets of behavioral interventions do not differ greatly for children and adolescents with autism. Even, booklet-delivered behavioral interventions specifically designed for autism have been developed and shown to be effective [17,18]. A behavioral sleep medicine toolkit outlining evaluation and treatment of insomnia in children with ASD has also been developed by the Autism Treatment Network (ATN) [19]. Furthermore, applied behavior analysis (ABA) and parent-based education have also shown improvement in sleep disorders in ASD [15].

Medications should be considered if behavioral interventions are ineffective or difficult to be implemented, especially in individuals with low-functioning autism, and should be used in combination with nonpharmacological strategies, which result in more sustained improvement [20]. Furthermore, medications should be initiated at the lowest effective dose and increased only if necessary. To decrease the risk of rebound insomnia, medication should be closely monitored and tapered gradually. Of particular concern, adolescents should be screened for alcohol and drug use, due to the additive effects of sedative hypnotic medications and recreational substances. Timing is crucial in the treatment of sleep disorders, as most hypnotic medications have their onset of action within 30 min and peak within 1–2 h. Therefore, dosing too early or too late would result in limited efficacy. Since over-the-counter (OTC) preparations are commonly used, an inquiry should be made to check concurrent use of OTC agents.

## 2. Melatonin

Melatonin is one of most common agents recommended for treatment of sleep difficulties and children with ASD. In a community survey of pediatricians, the majority had recommended a nonprescription medication for treatment of pediatric insomnia, with about 15% having recommended melatonin [16]. Melatonin is a hormone primarily synthesized in the pineal gland, with a major function to regulate circadian and core body temperature rhythms. Dim light melatonin onset (DLMO) is the single most accurate marker for assessing circadian pacemaker rhythms [21]. It is rapidly distributed to body tissues including the cerebrospinal fluid, bile, and saliva, where concentrations greatly exceed those found in the blood. Melatonin levels generally decline with advancing age. Its production is inhibited (by light) and is regulated by a circuitous pathway through the suprachiasmatic nuclei. Endogenous melatonin has both chrono biotic and sedative–hypnotic effects [22]. It has also been described that pharmacokinetic profiles of endogenous and supplemental melatonin in children with ASD are normal and comparable to typically developing children [23].

Several studies, including open-label data, uncontrolled trials to control trials, and meta-analysis, have been conducted in patients with ASD, establishing the efficacy of melatonin. Since it is considered a dietary supplement, its safety has not been thoroughly evaluated by the FDA. It is generally regarded as safe, despite lack of rigorous data pertaining to its use. Recommendations regarding use in children and adolescents are also mixed. National Institutes of Health (NIH) states that “Important questions remain about its usefulness, how much to take, when to take it, and its long-term safety” [24]. Because of its effects on other hormones, melatonin might interfere with development during adolescence. American Academy of Pediatrics states, “Melatonin appears to be effective in reducing time to sleep onset in adults (and, based on considerably less data, in children) for initial insomnia. This effect appears to last for days to weeks but not long-term. Thus, melatonin is not recommended for long-term use” [25]. By contrast, the Australian Sleep Health Foundation states that melatonin “may benefit children who are developing normally as well as children with Attention Deficit Hyperactivity Disorder, autism, other developmental disabilities or visual impairment” [26]. Here, we examine the efficacy studies regarding use of melatonin in children with autism spectrum disorders.

A key retrospective study, describing the use of melatonin in 107 subjects (2–18 years) utilizing a 3–6 mg dose, demonstrated that sleep concerns were no longer present for approximately 25% of the parents at follow-up after 1.8 years [27].

In another open-label trial, Malow and colleagues evaluated the efficacy of melatonin in 24 children aged 3–10 years, utilizing doses up to 9 mg. This study demonstrated that sleep onset latency was reduced by an average of 21.3 min (from 42.9 to 21.6 min, *p* < 0.0001), but there was no significant difference in total sleep duration, sleep efficiency, or night waking after 14 weeks of therapy. The study also demonstrated improvements in behavior and reduction in stereotypical and compulsive phenomena [28].

On the other hand, in another smaller placebo-controlled trial of 18 subjects, aged 2–15 years with ASD or fragile X syndrome, the melatonin group demonstrated significant improvements in total sleep duration and sleep onset time compared to the placebo group. Particularly, total sleep duration was increased by 21 min (*p* = 0.057) and sleep onset latency was decreased by 42 min (*p* = 0.017) [29].

There are limited data evaluating the efficacy of a controlled release formulation. In a double-blind crossover trial involving 51 subjects aged 2–18 years, where more than half of the subjects had ASD, melatonin 5 mg (1 and 4 mg immediate and controlled release, respectively) was found to reflect an increased mean total sleep duration (from 503.60 to 534.80 min, *p* < 0.01) and decreased sleep onset latency (65.18 to 32.48 min, *p* < 0.01) [30].

In a meta-analysis which included five randomized double-blind placebo-controlled crossover trials with 57 subjects with ASD and quantitative data sleep parameters, pooled data indicated that melatonin increased total sleep duration by an average of 73 min (*p* < 0.01) and decreased sleep onset latency by an average of 66 min (*p* < 0.001) [31]. However, there was no appreciable benefit for night wakings.

In another study, involving 146 children, using placebo-controlled intervention, 0.5–12, immediate release melatonin in a stepwise fashion, demonstrated only a small increase of 23 min in total sleep time but a much larger 38 min reduction in sleep latency. Additionally, melatonin had no demonstrable effect on night wakings [17].

Melatonin is available in different over-the-counter formulations ranging 1–10 milligrams in the United States and Canada, but in some countries, a prescription may be needed. Most commonly, a dose of 1–3 mg is recommended to be administered 30–60 min before intended bedtime [31]. However, if a circadian rhythm issue is identified, a lower dose (0.5–1 mg) administered earlier (3–4 h before bedtime) is recommended. An effective dose is not related to age or weight. Given the availability of different brands and the fact that strict regulations are not applicable to over-the-counter medications by the FDA, a concern is often raised regarding actual content of melatonin in the different formulations. In a recent study conducted in Ontario, which examined the melatonin content of different OTC formulations, actual melatonin content ranged from −83% to +478% of the labelled content. Furthermore, serotonin (5-hydroxytryptamine), a related indoleamine and controlled substance used in the treatment of several neurological disorders, was also identified in several supplements [32].

Melatonin is generally tolerated fairly well. Most studies published to date have not reported any serious safety concerns [28,29,33,34] (Table 1). Generally reported adverse effects include morning drowsiness, increased enuresis, headache, dizziness, and hypothermia. Some studies have reported suppression of the hypothalamic pituitary axis HPA axis with long term use and potential for precocious puberty on discontinuation [35]. Patients with a poor response have been shown to be poor CYP1A2 metabolizers.

Melatonin receptor agonists act selectively at MT1 and MT2 melatonin receptors and have demonstrated usefulness in children with autism spectrum disorder [36]. Ramelteon, the only drug in this class, has FDA approval for treatment of insomnia in adults. Side effects mainly include dizziness and fatigue and caution is advised for concomitant administration with Fluvoxamine.

## 3. Antipsychotics

Antipsychotics as a class have the largest body of evidence for treatment of behavioral difficulties, including aggression and irritability for treatment in autism spectrum disorder. These are usually prescribed for mood and behavioral comorbidities and secondarily have a beneficial effect for sleep. Antipsychotics have traditionally been categorized as typical and atypical antipsychotics. Typical antipsychotics, including haloperidol, fluphenazine, and thioridazine, are associated with higher incidence of extrapyramidal side effects and daytime somnolence. Newer, second-generation antipsychotics, including olanzapine, risperidone, and quetiapine, have a lower propensity for extrapyramidal side effects and are generally less sedating. There are limited efficacy and tolerability data for treatment of insomnia in children for this medication class. Few studies examining the effect on sleep architecture have shown that slow wave sleep is increased by olanzapine, ziprasidone, and Risperidone, whereas rapid eye movement (REM) suppression is greatest for ziprasidone and Risperidone [37]. Of the atypicals, risperidone and olanzapine have been prescribed for sleep disturbances in children [38] (Table 1). These agents are prescribed off label for treatment of insomnia and are not recommended to be prescribed routinely for this indication, especially as a first line pharmacotherapeutic agent. In particular, a guideline has been issued by the Canadian Academy of Child and Adolescent Psychiatry against its use for insomnia treatment in children, adults, or the elderly as a first line agent [39]. Other countries have also aimed to limit the number of prescriptions which may be allowed by government-subsidized programs.

## 4. Antidepressants

Limited data exist regarding use and efficacy of sedating antidepressants, selective serotonin reuptake inhibitors (SSRI) and tricyclic antideprssants (TCA), for treatment of sleep disturbances in children with autism spectrum disorder. These may be beneficial if insomnia is associated with comorbid psychiatric disorders. For example, sedating antidepressant such as mirtazapine and trazodone may be beneficial in a child with comorbid depression. These antidepressants promote sleep by antagonizing the effect of wake-promoting neurotransmitters, such as histamine, acetylcholine, noradrenaline, and serotonin. Most of these antidepressants suppress REM sleep and result in residual daytime sedation. Trazodone, in particular, is frequently preferred and used in psychiatric practice. Its efficacy has mainly been demonstrated in adults with psychiatric disorders (Table 1). Trazodone is a 5-HT2A/C antagonist and is one of the most sedating antidepressants with significant morning hangover effect. Fluoxetine, on the other hand, is generally associated with insomnia. The doses used for treatment of insomnia are generally lower compared to doses used for treatment of mood disorders. Selective serotonin reuptake inhibitors and tricyclic antidepressants can also be used if obsessional thoughts and anxiety significantly interfere with sleep onset. Amongst the TCAs, amitriptyline, imipramine, and doxepin are most sedating and used for treatment of insomnia in adults [40].

## 5. Anticonvulsants

Again, the data are limited regarding the use of anticonvulsants in the treatment of insomnia in this population group. Most of the trials have examined irritability and aggression and have reported improvement in these domains. The adverse events noted in these studies have ranged from insomnia to sedation [41]. Sedation is generally dose-related and tolerance is known to occur. The agents that have commonly been used in this category include valproate, lamotrigine, gabapentin, and carbamazepine. Gabapentin, in particular, has also demonstrated efficacy in adults with restless legs syndrome [42].

## 6. α-Adrenergic Agonists

The use of alpha agonist as off-label prescription has increased over the time. The prescription pattern in cohorts of children and adolescents (aged 4–18 years, *n* = 282,875) studied from 2009 to 2011 showed that about 68% of them received alpha agonists (shorter acting agents) as off-label medication for diagnosis of autism, primarily based on evidence from clinical trials without FDA approval. The study also revealed about 12% of them received alpha agonists for diagnosis of sleep and anxiety disorders without any evidence from randomized control trial in children [43].

Clonidine and guanfacine are the two primary alpha agonists used often as off-label medications for treating sleep disorder in autism.

Clonidine, an antihypertensive medication, is a central and peripheral a-adrenergic agonist which acts by stimulating presynaptic neurons, thereby decreasing noradrenergic release from the nerve terminals [44]. The two open-label retrospective studies in children and adolescents (aged 4–16 years) with autism and neurodevelopmental disorders documented that clonidine (dosing range: 0.05–0.225 mg/day) effectively improved sleep initiation and maintenance insomnia with good tolerability and few adverse events [45,46]. The potential side effects of clonidine include hypotension, bradycardia, irritability, dry mouth, and REM suppression, and its abrupt discontinuation can cause rebound hypertension and rebound REM [46,47].

Guanfacine, a selective α2A adrenergic receptor agonist, acts by stimulating postsynaptic alpha2A receptors in the prefrontal cortex (PFC) and in turn increases the noradrenergic transmission and connectivity of PFC networks [48]. Though immediate release guanfacine (dosing range: 0.5–2 mg/day) is frequently used off label for sleep disturbances in the pediatric population, there were no clinical trials conducted to determine its effectiveness [43,49,50]. Interestingly, a recent randomized, placebo-controlled trial of extended release guanfacine did not significantly improve sleep habits in Autism [51]. By contrast, a decrease in total sleep time was reported on polysomnography in placebo-controlled trial of extended release guanfacine [52].

## 7. Alzheimer’s Medications

Several Alzheimer’s medications have been studied for treating autism. Post mortem brain studies from individuals with autism have documented abnormalities in the cholinergic system and a connection between Alzheimer’s disease and autism has been proposed by many researchers [53,54]. One of the Alzheimer medications, donepezil, was found to be effective in improving behavioral and attention issues in autism.

Donepezil acts by selective reversible inhibition of acetylcholinesterase enzyme and increases cholinergic transmission in the synaptic cleft. In addition, in previous studies, donepezil was found to increase REM sleep in healthy and demented adults [55,56].

REM sleep is important for the promotion of cortical plasticity in developing brain. In children with autism, REM sleep abnormalities, such as immature organization, decreased quantity, and abnormal twitches, have been described [57]. In animal models, therapeutic augmentation of REM sleep has shown positive behaviors and improvement in learning [58]. Association of REM sleep augmentation and donepezil (dosing range: 1.25–5 mg/day) was studied in a small case series of children with autism (*n* = 5). This study demonstrated an increase in REM sleep percentage and decrease in REM latency with use of donepezil [59]. However, the findings of the study are limited due its small sample size. The potential side effects of donepezil include gastrointestinal symptoms such as nausea, vomiting, diarrhea, vivid dreams, insomnia, bradycardia, and hypotension [60].

## 8. Antihistamines

In the survey mailed to pediatricians (*n* = 671) by the American Academy of Pediatrics (AAP), antihistamines were found to be the most commonly reported nonprescription medication for sleep disorders [16]. Randomized controlled studies in typically developing children have documented improvement in transient insomnia with antihistamines. Diphenhydramine, a first-generation antihistamine, is prescribed often (dosing range: 0.5 mg/kg up to a maximum of 25 mg/day) by practitioners for sleep problems. It acts as a competitive histamine (H1) receptor blocker in the central and peripheral nervous system, causing a sedative and hypnotic effect. Another H1 receptor antagonist, Trimeprazine (dosing range: 45–90 mg/day), has also been shown to improve nocturnal awakenings in children with chronic sleep disturbances. The potential side effects include sedation and anticholinergic effects, including fever, blurred vision, dry mouth, constipation, urinary retention, tachycardia, and confusion [48,49]. Despite widespread use of antihistamines, the clinical trials in patients with autism spectrum disorder have been limited. A European open-label study documented niaprazine, a piperazine derivative, which also acts as an antihistamine (dosing range: 1 mg/kg/day three times daily), to be effective in improving sleep problems in children with autism and with mild to moderate intellectual disability. Niaprazine has not been approved for use in the United States [61].

## 9. Sedative and Hypnotics

Benzodiazepines (BZDs) are frequently prescribed in adults with insomnia. However, they are prescribed less frequently in the pediatric population because of their side-effects profile includes sedation, headaches, dizziness, cognitive impairment, rebound insomnia, and physical and behavioral dependence. There have only been limited studies in pediatrics, which have shown improvement in sleep disorders with use of BZDs.

The mechanism of action of BZDs is primarily to bind to α and γ subunits of the gamma aminobutyric acid (GABA) chloride receptor, inducing a conformational change in the receptor complex, and facilitating GABA. The inhibitory action of GABA on the central nervous system causes sedative, anxiolytic, and muscle-relaxing effects [62]. Because of BZDs’ muscle-relaxant property, they should be used cautiously in children with a sleep-related breathing disorder. Benzodiazepines are also well known to alter normal sleep state architecture and there has been polysomnographic evidence of atypical sleep spindles and slow wave sleep suppression with chronic BZD use.

The only benzodiazepine studied in sleep disorders in children with Autism was clonazepam. Clonazepam, an intermediate acting BZD, was found to be effective in treating parasomnias, partial arousals, periodic limb movement disorder, and nocturnal biting in children with developmental disabilities [49]. Its half-life ranges between 18–39 h, with time to peak level of 1–2 h, and it has a primary renal elimination [62]. A small case study of children (*n* = 11) with autism and REM behavior disorder (RBD) found clonazepam (dosing range: 0.25–0.5 mg/day) to be effective in improving sleep disturbances. It was well tolerated in most of the participants, except one paradoxical response in a child [57].

The nonbenzodiazepines, zaleplon, zolpidem, and eszopiclone, often called as ‘Z-drugs’, have similar pharmacology to BZDs, but are not chemically related to BZD. They act at benzodiazepine-1 subtype in the GABA receptor complex. All Z-drugs have a relatively short life. In contrast to BZD, they do not cause significant residual daytime sedation, cognitive, or memory impairment [63]. In addition, they do not typically cause rebound insomnia (an exacerbation of insomnia on abrupt cessation of a hypnotic) with abrupt discontinuation, which is one of the worsening adverse effects of BZD. Zaleplon and zolpidem have been used in children but data on the use of eszopiclone are limited [47]. Clearance of nonbenzodiazepine receptor agonist drugs in children is three times higher than in adults, which can cause medication ineffectiveness and may even lead to terrifying sleep states, like sleepwalking and sleep-related hallucinations [47]. There were no clinical trials available for this category of medications in autism.

## 10. Oral Iron Supplement

Serum ferritin, a storage form of iron (level below 50 ng/mL), was associated with restless legs syndrome (RLS) [64]. In a retrospective chart review study of children with autism spectrum disorder (*n* = 9791), significantly low serum ferritin levels were identified and associated with several sleep disorders, including periodic limb movements of sleep (27 ng/mL), sleep fragmentations (24 ng/mL), and poor sleep efficiency (7 ng/mL) [13].

Iron deficiency states were documented in the pathophysiology of RLS and the severity of iron deficiency states was correlated with the severity of RLS. Iron plays a major role in the dopamine production pathway; it acts as a cofactor for a rate limiting enzyme tyrosine hydroxylase in the dopamine production cycle. In patients with RLS, low cerebrospinal iron levels and low iron in the substantia nigra on magnetic resonance imaging were noted. Iron supplementation was found to be effective in the treatment of low ferritin with sleep disorders. The RLS foundation medical advisory board recommends iron therapy for low ferritin level below 50 ng/mL [64].

An open-label trial of oral iron supplement (6 mg elemental iron/kg/day) for 8 weeks in children with autism showed improvement in the sleep disturbance scale with an increase in serum ferritin level [65]. Potential side effects of oral iron include metallic taste, vomiting, nausea, constipation, diarrhea, and black/green stools [66].

## 11. Conclusions

In summary, sleep difficulties frequently co-occur in children with autism spectrum disorder and medications are often prescribed. However, limited evidence exists regarding the use and efficacy of medications for the treatment of sleep disorders in this population group. Melatonin has demonstrated good efficacy in open-label and placebo-controlled trials; however, long term effects still need to be thoroughly explored. Despite evidence and widespread use of medications to treat sleep disorders in this population group, no medications are FDA-approved for this indication. Identification and management of psychiatric comorbidities is important to achieve favorable outcomes. There is a need for more information and protocols needed to guide management for this population group.

## Figures and Tables

**Table 1 medsci-06-00095-t001:** Selected medications for sleep referenced in the publications on autism spectrum disorder (ASD) and psychiatric comorbidities.

Medication	Dosing Range	Common Side Effects	Clinical Use
Melatonin	1–3 mg	Nausea, headache, dizziness, hypothermia	Effective in children with developmental disorders, Autism spectrum disorders, Jet Lag
**Antipsychotics** Olanzapine Risperidone	2.5–10 mg 0.25–2 mg	Daytime drowsiness, weight gain, hypercholesterolemia, diabetes Mellitus, prolactin elevation	Effective in comorbid maladaptive behaviors including irritability, aggression and self-injury
**Antidepressants** Trazodone *	20–50 mg (adult dose)	Dizziness, morning drowsiness, priapism, hypotension	Useful with comorbid depression
**Alpha-adrenergic agonist** Clonidine (immediate release) Guanfacine (immediate release)	0.05–0.225 mg 0.5–2 mg	Hypotension, bradycardia, irritability, dry mouth, REM suppression. Abrupt discontinuation causing rebound hypertension and rebound REM	Sleep initiation and maintenance insomnia
**Antihistamine** Diphenhydramine Trimeprazine Niaprazine (not approved for use in USA)	0.5 mg/kg up to 25 mg 45–90 mg 1 mg/kg/day three times daily	Sedation and anticholinergic effects, including fever, blurred vision, dry mouth, constipation, urinary retention, tachycardia and confusion	Transient insomnia
**Sedative and Hypnotics** Clonazepam	0.25–0.5 mg	Sedation, headaches, dizziness, cognitive impairment, rebound insomnia, physical and behavioral dependence	Parasomnias, periodic limb movement disorder, nocturnal biting
**Iron supplements** Oral Iron	6 mg elemental iron/kg/day	Metallic taste, vomiting, nausea, constipation, diarrhea, black/green stools	Sleep disturbances, Periodic limb movements of sleep, restless legs syndrome
**Alzheimer’s medications** Donepezil	1.25–5 mg	Gastrointestinal symptoms, vivid dreams, insomnia, bradycardia, hypotension	Increased REM sleep percentage and decreased REM latency

* Adult data. REM: Rapid eye movement.

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
