# Peer review of "Pharmacological Approach to Sleep Disturbances in Autism Spectrum Disorders with Psychiatric Comorbidities: A Literature Review"

_medsci, 2018, doi:10.3390/medsci6040095_

Round 1

Reviewer 1 Report

Authors have provided a comprehensive overview of medications for sleep in children with autism. The detailed review of melatonin and antihistamines would be appreciated by readers.

Authors may want to comment on the recent trend of using Seroquel/Quetiapine for indication of sleep leading to Medicaid and health plans including an edit to restrict the use of the drug for sleep in children.

Overall, it is a timely update for readers.

Author Response

Response to Reviewer 1

Authors have provided a comprehensive overview of medications for sleep in children with autism. The detailed review of melatonin and antihistamines would be appreciated by readers.

Authors may want to comment on the recent trend of using Seroquel/Quetiapine for indication of sleep leading to Medicaid and health plans including an edit to restrict the use of the drug for sleep in children.

Overall, it is a timely update for readers

Thank you for the feedback. We have elaborated on the recent concern for use of antipsychotics for sleep and have outlined guidelines issued by different organizations advocating against its use as a first line agent.

Reviewer 2 Report

*Introduction*

The degree of impairment among individuals with ASD is variable, thereby requiring the distinction between individuals with low-functioning autism and high-functioning autism, defined as those which have an intellectual quotient that is below average (<70) and above average (≥70), respectively. What the current DSM-V fails to capture is that individuals with low-functioning autism experience significantly graver impairments than those experienced by their higher functioning counterparts. Thus the main approach to treatment for sleep issues in autism should be behavioural based (such as cognitive behaviour therapy, bright light therapy, sleep hygiene). However in low functioning autism, where behavioural based interventions are difficult or unable to be implemented, pharmacotherapy is the next line of treatment. I think there needs to be a more sophisticated introduction outlining that pharmacological interventions should be used as a second line of treatment and this will most often be an approach required for individuals with LFA.

*Description of aetiology of sleep problems in Autism*

This is far too simplistic. It is well established that ASD is characterised anxiety, sensory sensitivities, problems with regulating arousal etc – all of which impact on sleep, arousal thresholds and sleep onset latency. I suggest the authors consider papers such as Cohen et al. /The relationship between sleep and behavior in autism spectrum disorder (ASD): a review./ J Neurodev Disord. 2014;6(1):44.

*Section on Melatonin*

This section needs to be more balanced. It is very biased toward promoting the use of Melatonin and its analogues which I personally believe is very problematic. It should be noted that bright light therapy should be considered in treating chronbiological issues in children as a first line of treatment before considering Melatonin. Melatonin should also be strictly used as a chronobiotic in small doses when a circadian timing issue has been identified. It should not be used in larger doses as a hypnotic. I suggest the authors read more widely to incorporate both the pros and cons of prescribing Melatonin to children (e.g. Kennaway DJ /Potential safety issues in the use of the hormone melatonin in paediatrics/ Journal of Paediatrics and Child Health 51 (2015) 584–589)).

The authors then review a range of pharmacological treatments for adults which aren’t really relevant for children and that’s what they conclude in most instances so I’m not sure what value this adds. I think it would be better if they added more depth to the coverage of treatments that are more realistically considered as a treatments for children. 

Author Response

Dear Reviewers, 

We would like to thank you for considering our manuscript and also for your time, effort and insightful comments. We thank the reviewers for recognizing the strengths of the manuscript. In our attempt to address the critiques, please find below an itemized list of our efforts to address the reviewer concerns, which are in bold to clearly distinguish from the reviewers’ comments. 

REVIEWER 2:

 Remarks:

*Introduction*

The degree of impairment among individuals with ASD is variable, thereby requiring the distinction between individuals with low-functioning autism and high-functioning autism, defined as those which have an intellectual quotient that is below average (<70) and above average (≥70), respectively. What the current DSM-V fails to capture is that individuals with low-functioning autism experience significantly graver impairments than those experienced by their higher functioning counterparts. Thus the main approach to treatment for sleep issues in autism should be behavioural based (such as cognitive behaviour therapy, bright light therapy, sleep hygiene). However in low functioning autism, where behavioural based interventions are difficult or unable to be implemented, pharmacotherapy is the next line of treatment. I think there needs to be a more sophisticated introduction outlining that pharmacological interventions should be used as a second line of treatment and this will most often be an approach required for individuals with LFA.

We discussed about high functioning vs low functioning autism in the introduction and elaborated on the treatment modalities for sleep disorders in autism spectrum disorders and have emphasized the role of behavioral interventions and that behavioral interventions should be considered first prior to initiating pharmacotherapy and should be used in combination with pharmacotherapy for sustained effects.

*Description of aetiology of sleep problems in Autism*

This is far too simplistic. It is well established that ASD is characterised anxiety, sensory sensitivities, problems with regulating arousal etc – all of which impact on sleep, arousal thresholds and sleep onset latency. I suggest the authors consider papers such as Cohen et al. /The relationship between sleep and behavior in autism spectrum disorder (ASD): a review./ J Neurodev Disord. 2014;6(1):44.

Thanks for suggesting this article We included this reference and discussed about low functioning autism, variablities in presentation. We discussed about medical and pscyhiatric co morbidities such as ADHD, Anxiety and  their influence on sleep disturbances in ASD. 

*Section on Melatonin*

This section needs to be more balanced. It is very biased toward promoting the use of Melatonin and its analogues which I personally believe is very problematic. It should be noted that bright light therapy should be considered in treating chronbiological issues in children as a first line of treatment before considering Melatonin. Melatonin should also be strictly used as a chronobiotic in small doses when a circadian timing issue has been identified. It should not be used in larger doses as a hypnotic. I suggest the authors read more widely to incorporate both the pros and cons of prescribing Melatonin to children (e.g. Kennaway DJ /Potential safety issues in the use of the hormone melatonin in paediatrics/ Journal of Paediatrics and Child Health 51 (2015) 584–589)).

The authors then review a range of pharmacological treatments for adults which aren’t really relevant for children and that’s what they conclude in most instances so I’m not sure what value this adds. I think it would be better if they added more depth to the coverage of treatments that are more realistically considered as a treatments for children. 

We have elaborated on this concern in the section on Melatonin in order to present a more balanced view of current evidence and recommendations. In particular, we have included the caveat that evidence regarding efficacy of melatonin in children is limited at best and there is lack of universal guidelines regarding it use, especially regarding long term use.

We have also abbreviated the section on sedatives and hypnotics to highlight only the evidence in children and adolescents especially in children with autism. We have continued to include psychotropics such as Trazodone which are prescribed often in adolescents and may therefore be useful to the practicing clinician. In this case, we have highlighted that data for its use is existent primarily in adults.
